# Disrupting Model Training with Adversarial Shortcuts

**Ivan Evtimov** [1]  **Ian Covert** [1]  **Aditya Kusupati** [1]  **Tadayoshi Kohno** [1]

## Abstract

When data is publicly released for human consumption, it is unclear how to prevent its unauthorized usage for machine learning purposes. Successful model training may be preventable with carefully designed dataset modifications, and we present a proof-of-concept approach for the image classification setting. We propose methods based on the notion of *adversarial shortcuts*, which encourage models to rely on non-robust signals rather than semantic features, and our experiments demonstrate that these measures successfully prevent deep learning models from achieving high accuracy on real, unmodified data examples.

## 1. Introduction

Datasets are publicly released with a diverse set of use cases (e.g., posting images for friends, promoting photography work), but not all stakeholders will consent to the data's usage for machine learning (ML). Different parties may desire that their copyright be respected, or they may wish to avoid potentially harmful uses such as deepfakes, facial recognition systems, or other biometric models.

To help manage such situations, we consider the problem of modifying datasets to ensure that they are unusable for ML purposes. By developing a disruptive modification that preserves the data's original semantics, we hope to provide an orthogonal approach to traditional privacy preservation avenues such as anti-scraping technology and legal agreements, which in recent years have proved insufficient at protecting user data (Hill et al., 2020).

Because this is a broad and challenging problem, our initial focus is the canonical setting of multi-class image classification. Our goal is to modify a clean dataset so that ML models, and primarily deep neural networks (DNNs), achieve

---

[1]Paul G. Allen School of Computer Science & Engineering, University of Washington. Correspondence to: Ivan Evtimov <ie5@cs.washington.edu>.

*Accepted by the ICML 2021 workshop on A Blessing in Disguise: The Prospects and Perils of Adversarial Machine Learning. Copyright 2021 by the author(s).*

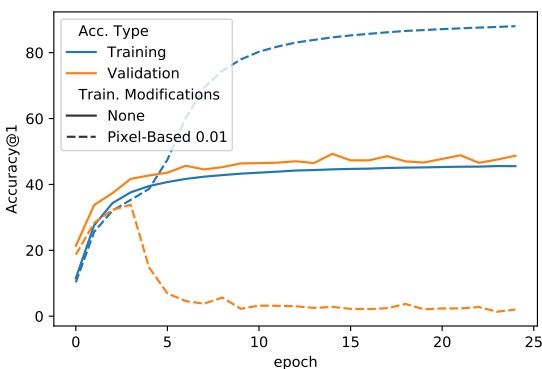

*Figure 1.* ResNet18 train and validation accuracy on ImageNet protected with the pixel-based modification at $\mu = 0.01$, compared with unmodified ImageNet. Validation accuracy can reach up to 70% with unmodified data, but our modified dataset prevents effective learning within the first several epochs.

high training accuracy while failing to generalize to unmodified test examples. As a mechanism for such modifications, we explore *adversarial shortcuts*, a method that encourages DNNs to lazily rely on spurious signals rather than robust, semantic features. While recent work focuses on the perils of shortcuts in deep learning (DeGrave et al., 2021), we identify a potential use-case as a security measure.

Our contributions in this work are as follows:

- We introduce the notion of adversarial shortcuts and propose three dataset modification techniques to prevent DNNs from learning useful classification functions.

- We evaluate each technique on the popular CIFAR-10 (Krizhevsky et al.) and ImageNet (Russakovsky et al., 2015) datasets and find that the proposed techniques severely limit the test set accuracy of state-of-the-art models. We also verify that our techniques are robust to certain simple countermeasures.

- We compare our approach to a concurrent proposal (Fowl et al., 2021a) and show that our simpler approach based on adversarial shortcuts proves more effective at disrupting model training.

## 2. Related Work

Our work focuses on making datasets unusable for training ML models. Concurrent work that considers the same problem (Fowl et al., 2021a) proposed a poisoning approach based on second-order gradients. Similarly, Shumailov et al. (2021) proposed a method to disrupt training with a data ordering attack, but our setting is more challenging in that we do not assume control over model optimization. Other closely related research directions are trojaning and backdooring (Schwarzschild et al., 2020; Goldblum et al., 2020), where the adversary can influence the inputs at both training and inference time, and in some cases the training procedure. Data poisoning (Biggio et al., 2011; 2012) is another direction where subtle changes to the classification's decision boundary are induced by modifying a few training examples. Targeted data poisoning attacks focus on influencing model behavior on specific inference examples, either in a transfer learning setting (Shafahi et al., 2018; Zhu et al., 2019; Aghakhani et al., 2020) or when training from scratch (Muñoz-González et al., 2017; 2019; Huang et al., 2020a; Geiping et al., 2020; Huang et al., 2021; Fowl et al., 2021b)

We explore shortcuts as a mechanism to disrupt model training, an idea with some precedent in prior work. Research on the learning tendencies of DNNs has found that conventionally trained models often rely on non-robust, localized, texture-based features (Zhang et al., 2016; Jo & Bengio, 2017; Madry et al., 2017; Geirhos et al., 2018; Ilyas et al., 2019), and, when available, confounders or shortcuts (Zech et al., 2018; Badgeley et al., 2019; DeGrave et al., 2021). Rather than attributing model failures to naturally occurring shortcuts, we purposely introduce shortcuts to discourage a model's reliance on robust, semantic features.

Finally, our aim is the reverse of instance hiding (Huang et al., 2020b; Carlini et al., 2020), which tries to create a dataset that is useful for ML but uninterpretable to humans, and our approach is similar in spirit to image watermarking (Podilchuk & Delp, 2001; Singh & Chadha, 2013; Dekel et al., 2017), which aims to prevent unauthorized usage of publicly released image data.

## 3. Setup and Goals

Assume that we have a dataset $\mathcal{D}_{\text{train}} = \{(x_i, y_i)\}_{i=1}^{n}$ where $x_i \in \mathbb{R}^{w \times h \times c}$ are RGB images and $y_i \in \{1, ..., K\}$ are the corresponding labels for a classification task. $\mathcal{D}_{\text{train}}$ is assumed to be drawn from a data generating distribution $\mathcal{D}$, and as in standard supervised learning, we assume that models trained on $\mathcal{D}_{\text{train}}$ are used to classify test samples from $\mathcal{D}_{\text{test}}$, which is drawn independently from $\mathcal{D}$.

Rather than releasing $\mathcal{D}_{\text{train}}$ directly, which can be used to train a model that achieves high accuracy on $\mathcal{D}_{\text{test}}$, our goal is to create a modified dataset $\mathcal{D}'_{\text{train}} = \{(x'_i, y_i)\}_{i=1}^{n}$ with

the following properties:

- **Semantics in $\mathcal{D}'_{\text{train}}$ are preserved.** The modified images $x'_i$ should differ from the original images $x_i$ minimally, ideally being visually indistinguishable, but at least retaining the important semantics (objects, shapes, colors, etc). The labels $y_i$ are unmodified, and we assume that a party obtaining our modified dataset may reconstruct $y_i$ even if labels are not provided.

- **Models trained on $\mathcal{D}'_{\text{train}}$ achieve low accuracy on $\mathcal{D}_{\text{test}}$.** When DNNs are trained to predict the labels $y_i$ given images $x'_i$ from the modified dataset, the models should be unable to generalize to unmodified examples, ensuring low accuracy on the test dataset $\mathcal{D}_{\text{test}}$.

## 4. Protective Dataset Modifications

In this section, we introduce dataset modifications that encourage DNNs to rely on spurious signals rather than robust, generalizable features. We propose three approaches: a sparse pixel-based pattern, a visible watermark, and a brightness modulation. All three generate modifications that are unique to each class $k \in \{1, \ldots, K\}$, creating a shortcut that the DNN can use to quickly achieve high accuracy on the training data while failing to generalize to unmodified examples. We refer to such modifications as *adversarial shortcuts*, and each technique is tuneable, allowing us to control the tradeoff between disrupting training and preserving visual features in the data.

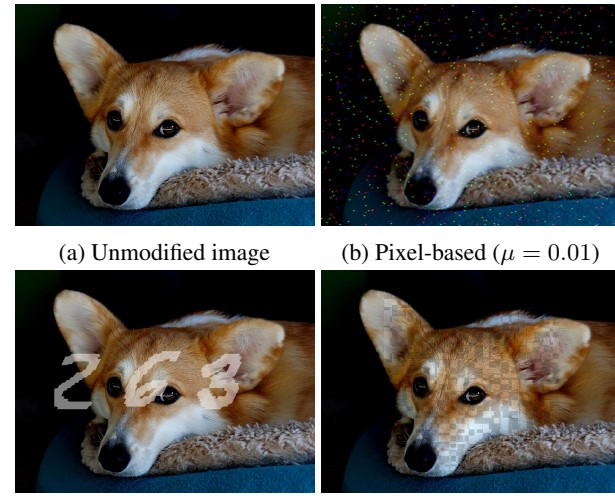

(a) Unmodified image     (b) Pixel-based ($\mu = 0.01$)

(c) Visual watermark ($\alpha = 0.5$) (d) Brightness mod. ($\gamma = 0.9$)

*Figure 2.* Demonstration of our dataset modification techniques. The image depicted here is available at https://www.flickr.com/photos/volvob12b/9797687423, was accessed on June 3, 2021, and is distributed in the public domain. This would have class index 263 for "Pembroke, Pembroke Welsh corgi."

### 4.1. Sparse pixel-based patterns

The first approach we introduce is a sparse, pixel-based modification. We generate random perturbation masks $\Delta_k \in \{0,1\}^{w \times h \times c}$ for each class $k \in \{1, \dots, K\}$ with entries determined as follows: for each value, we sample $\delta \sim \mathcal{N}(\mu, \sigma)$ and set the value to one if $\delta$ exceeds the middle of the pixel brightness range (e.g., 0.5 with 0-1 normalization), and zero otherwise. In practice, we fix $\sigma = 0.2$ and experiment with different $\mu$ values.

With the masks fixed, we then modify images in the pixels indicated by their corresponding perturbation masks. Assuming that the maximum pixel value in the dataset is given by $x_{\max} \in \mathbb{R}$, we generate the modified image $x_i'$ with label $y_i = k$ using the formula

$$x_i' = (1 - \Delta_k) \odot x_i + \Delta_k \cdot x_{\max}. \tag{1}$$

An example of this perturbation is shown in Figure 2, where a small portion of pixels are set to the maximum brightness.

### 4.2. Visible watermarks

The next approach we introduce is a visible, class-specific watermark. If the watermark is prominent and easy to detect, a DNN can use it as a shortcut to achieve high accuracy without relying on robust features. To generate shapes with a sufficient degree of variation, which is known to make watermarks more difficult to remove (Dekel et al., 2017), we create watermarks by enumerating the class indices using digits from the MNIST dataset (LeCun, 1998).

For example, in CIFAR-10 (Krizhevsky et al.) the "airplane" class has index 0, so we create a watermark for each airplane example by randomly sampling a zero from the MNIST dataset. For ImageNet, which has 1000 classes, the watermarks require up to three randomly selected digits. The watermark generation process for each class $k \in \{1, \dots, K\}$ can be understood as sampling a binary image $M \in \{0,1\}^{w \times h \times c}$ from a random variable $\mathcal{M}(k)$, which we then blend with the original image $x_i$ using a parameter $\alpha \in [0, 1]$ as follows:

$$x_i' = \alpha \cdot M + (1 - \alpha) \cdot M \cdot x_i + (1 - M) \cdot x_i. \tag{2}$$

The blending parameter $\alpha$ controls how visible the watermark is, with $\alpha = 0$ having no effect and $\alpha = 1$ overlaying the watermark on the original image. An example with $\alpha = 0.5$ is shown in Figure 2, with index 263 for the "Pembroke, Pembroke Welsh corgi" class.

### 4.3. Brightness modulation

While the two previous approaches provide shortcuts that can successfully disrupt model training, they may prove easy to remove with basic countermeasures. Our next approach is designed to be more difficult to circumvent. Rather than creating a localized, visually distinguishable perturbation, we now modify images using a randomized brightness modulation that either brightens or darkens pixels identically for images in each class.

The brightness modulation for each class is generated as follows. At the start, we randomly sample a location in the image that serves as the center of a square; we then decide, with equal probability, whether to darken or brighten the corresponding pixels. Given a parameter $\gamma \in [0.5, 1]$, we darken pixels by multiplying them by $\gamma$ or brighten them by multiplying by $2 - \gamma$. We perform $T$ iterations of this process with $T$ distinct squares, which can and do overlap, resulting in a checkerboard-type pattern.

This process is equivalent to sampling a class-specific mask $B_k \in \mathbb{R}^{w \times h \times c}$, where an image $x_i$ with class $y_i = k$ is modified using the following formula:

$$x_i' = B_k \odot x_i. \tag{3}$$

An example of this modification is shown in Figure 2 with the parameter $\gamma = 0.9$ and $T = 600$ iterations. For all experiments with ImageNet we set $T = 600$, and for CIFAR-10 we use $T = 32$.

## 5. Evaluation

In this section, we evaluate our proposed techniques for disrupting model training; see Appendix A for more details on the setup. Although we do not reach state-of-the-art training accuracy on CIFAR-10 and ImageNet, either due to computational constraints or insufficient hyperparameter tuning, we ensure a fair comparison by using identical training procedures across all experiments.

### 5.1. Training on modified CIFAR-10

We first test our dataset modifications on CIFAR-10. Figure 3 summarizes the results from training a ResNet18 architecture with various dataset modifications. Figures 6, 7, and 8 provide more ablations and details, including different model architectures and more parameter settings for each adversarial shortcut. The best achievable validation accuracy after 50 epochs with the unmodified version of CIFAR-10 is above 70% accuracy, while all of our modifications, even at the weakest settings we tested, have a significant impact on model performance.

Relatively small pixel-based perturbations are enough to nearly halve the accuracy: with $\mu = 0.01$, CIFAR-10 classifiers achieve at best no more than 40% accuracy. This setting corresponds to modifying only 22 out of 3,072 pix-

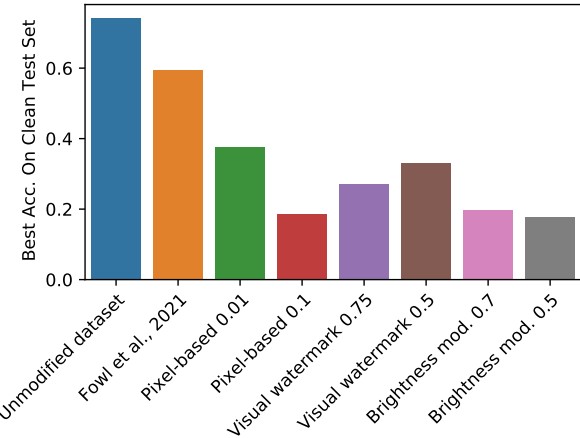

Figure 3. Best achievable test accuracy after 50 epochs when training ResNet18 on CIFAR-10 with different dataset modifications.

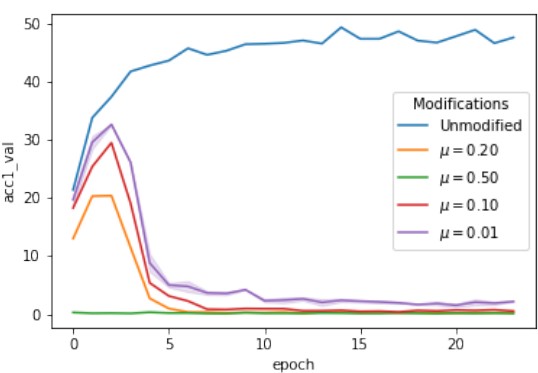

Figure 4. Validation accuracy when training ResNet18 on ImageNet at different levels of sparse pixel-based pattern protections.

els, on average. Similarly, visible watermark perturbations with a blend factor of $\alpha = 0.5$ and brightness modulations with parameter $\gamma = 0.9$ succeed in reducing the validation accuracy to less than 40%.

Fowl et al. (2021a) shared a version of CIFAR-10 protected with their proposed approach, which we compared with our methods.[1] While the validation accuracy does not reach 70%, as with training on the unprotected version of the dataset, models trained on the dataset with Fowl et al. (2021a) protections manage to achieve up to 60% accuracy, which is significantly higher than our methods (also see Figure 9).

We also tested the stability of our proposed modifications for disrupting training when certain countermeasures are in place. For this purpose, we considered two categories of countermeasures: aggressive training set augmentations and the addition of Gaussian noise. Results for the pixel-based approach with these countermeasures are shown in Figures 11, 12 and 13. Our findings generalize, and the best achievable accuracy across the same set of hyperparameters and random seeds remains the same. However, for the brightness modulation method with $\gamma = 0.9$, aggressive augmentations are effective at undoing the modifications and allowing effective training (see Figure 13). We suggest using a stronger setting of $\gamma = 0.70$ that makes the brightness modulation more visible.

### 5.2. Training on modified ImageNet

Next, we perform experiments on ImageNet. Although we do not have the computational resources to train with a va-

riety of hyperparameter and random seed choices, several takeaways are apparent from Figure 4 and additional results in Appendix A (Figure 5). First, sparse pixel-based pattern protections and visible watermark protections remain effective. In both cases, the best achievable validation accuracy is less than 30%, whereas training on the unprotected version of ImageNet easily achieves more than 50% accuracy with the same setup. This is again achievable with fairly minor modifications, such as pixel-based with $\mu = 0.01$ and visible watermarking with $\alpha = 0.5$.

Furthermore, the plots of accuracy on the clean validation set and the protected training set in Figure 1 reveal an interesting dynamic. While the model can fit the training set extremely well, achieving up to 90% accuracy, it does not generalize to the validation set. This suggests that our objective of disrupting training with a non-robust shortcut is successful, and that the models fits the simple class-specific pattern as opposed to the true semantics. This divergence in training and validation accuracy, or the rapid increase in the generalization gap, does not manifest when training with the unmodified ImageNet data (Figure 1).

## 6. Discussion

Our experiments show that it is possible to disrupt DNN training by modifying datasets with simple patterns, such as our adversarial shortcuts, that discourage models from relying on robust, generalizable features. These modifications can reduce model accuracy on clean data while having minimal impact on the image semantics. Our work focuses on the narrow setting of multi-class image classification, but there is great potential for future work that considers more effective dataset modifications, attempts to undo protective modifications, and develops new approaches for different ML tasks, e.g., preventing the unauthorized development of deepfakes or facial recognition systems.

---

[1] The version of their defense that the authors shared with us for this test has parameters $\epsilon = 8/255$, but we note that stronger training disruptions may be achieved with different parameters.

## Acknowledgements

The authors would like to thank Gabriel Ilharco, Kaiming Cheng, and Samuel Ainsworth for helpful discussions on the subject matter and for their thoughtful feedback.

This work was supported in part by the University of Washington Tech Policy Lab, which receives support from: the William and Flora Hewlett Foundation, the John D. and Catherine T. MacArthur Foundation, Microsoft, the Pierre and Pamela Omidyar Fund at the Silicon Valley Community Foundation; it was also supported by the US National Science Foundation (Award 156525).

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

## A. Experimental Setup

We experiment with two standard image classification datasets: CIFAR-10 (Krizhevsky et al.) and ImageNet (Deng et al., 2009). In both cases, we apply our protective modifications to the training set and keep the validation set intact. By measuring accuracy on the validation set, we can observe how well the trained models solve their intended classification task. If our protections are successful, the best achievable accuracy should be considerably lower than when training on an unprotected dataset.

For the experiments with CIFAR-10, we run training for 50 epochs at a batch size of 1024 and vary the learning rate, the model architecture and the random seed. Specifically, we experiment with learning rates 0.1, 0.01, 0.001, 0.0001; with the ResNet18 (He et al., 2016), DenseNet201 (Huang et al., 2017), VGG11 (Simonyan & Zisserman, 2014), and SqueezeNet (Iandola et al., 2016) architectures; and with seeds 3525462, 15254521, 63246662, 32542462. We then report the best achievable accuracy across all the runs for several different settings of each of our intensity parameters ($\mu$ for pixel-based patterns, $\alpha$ for watermarks, and $\gamma$ for brightness modulations). All models are as implemented

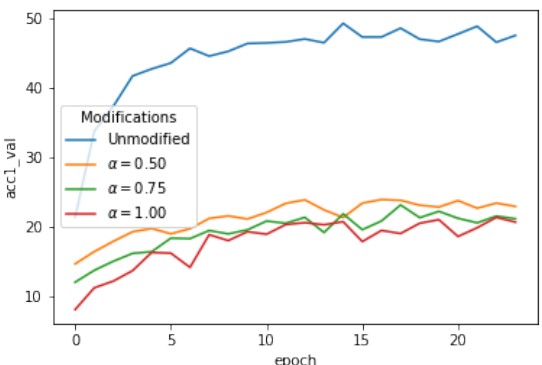

Figure 5. Validation accuracy when training ResNet18 on ImageNet at different levels of visible watermarking protections.

in torchvision (Paszke et al., 2019) and are trained from scratch.

For ImageNet, we use the training script available at https://github.com/pytorch/examples/tree/master/imagenet and the ResNet18 (He et al., 2016) architecture. We only train using the default hyperparameter values due to our limited computational resources.

For testing the stability of our approach with aggressive augmentations, we employ random cropping of 28 by 28 images, random horizontal flips, random Affine transformations (translation by up to 0.1 and rotation between -30 and 30 degrees), random color jitter (brightness adjustment by up to 0.8 and contrast adjustment between 0.9 and 1.08). Each augmentation is sampled at random for each new training image and for each epoch during training. When testing the stability of our approach with Gaussian noise, we add noise with mean 0.0 and $\sigma = 0.05$.

## B. Additional Figures

Here, we include figures that do not fit in the page limit.

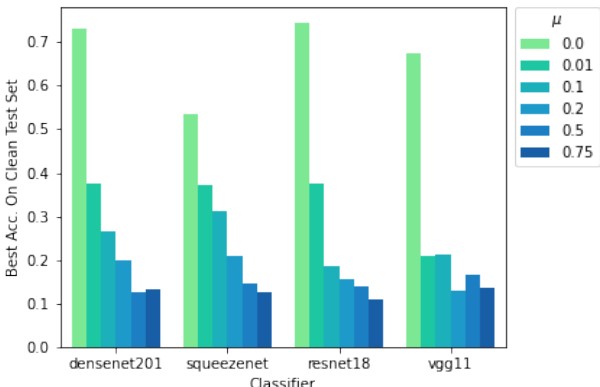

Figure 6. Best achievable accuracy after 50 epochs across a range of hyperparameters and random seeds for the pixel-based modification. $\mu = 0$ corresponds to the unmodified, clean dataset for this approach.

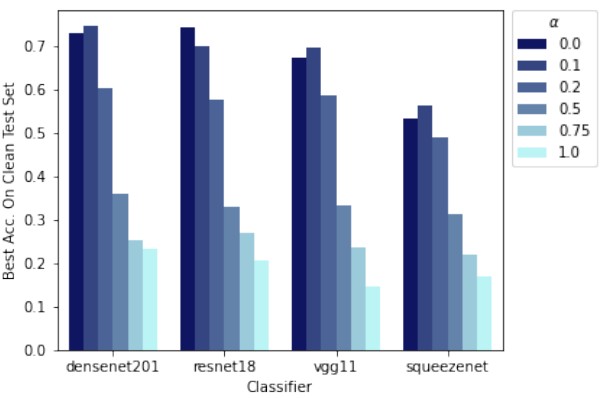

Figure 7. Best achievable accuracy after 50 epochs across a range of hyperparameters and random seeds for the visible watermark approach. $\alpha = 0$ corresponds to the unmodified, clean dataset for this approach.

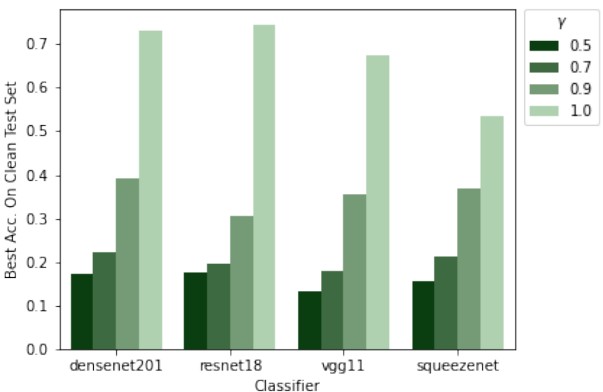

Figure 8. Best achievable accuracy after 50 epochs across a range of hyperparameters and random seeds for the brightness modulation approach. $\gamma = 1.0$ corresponds to the unmodified, clean dataset for this approach.

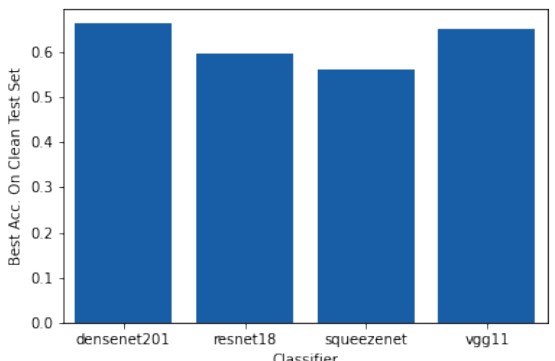

Figure 9. Best achievable accuracy after 50 epochs across a range of hyperparameters and random seeds for the Fowl et al. (2021a) approach

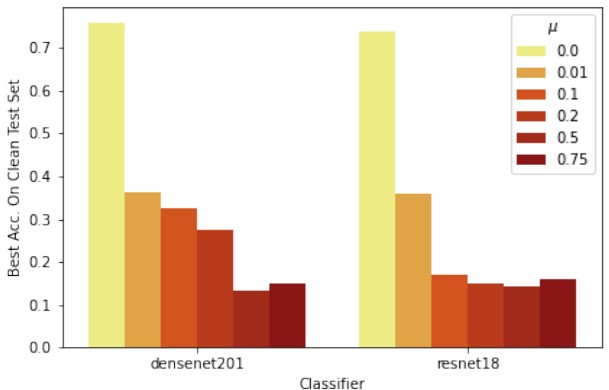

Figure 10. Best achievable accuracy after 50 epochs across a range of hyperparameters and random seeds for the pixel-based approach after applying Gaussian noise to the images. $\mu = 0$ corresponds to the unmodified, clean dataset for this approach.

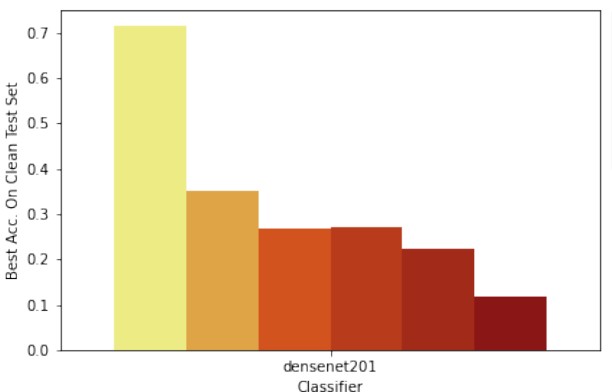

Figure 11. Best achievable accuracy after 50 epochs across a range of hyperparameters and random seeds for the pixel-based approach after applying aggressive training-time augmentations. $\mu = 0$ corresponds to the unmodified, clean dataset for this approach.

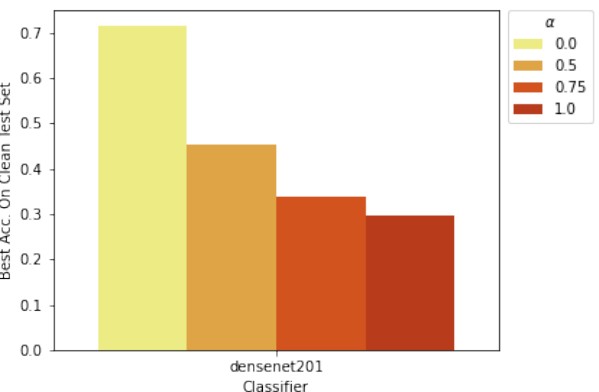

Figure 12. Best achievable accuracy after 50 epochs across a range of hyperparameters and random seeds for the visible watermark approach after applying aggressive training-time augmentations. $\alpha = 0$ corresponds to the unmodified, clean dataset for this approach.

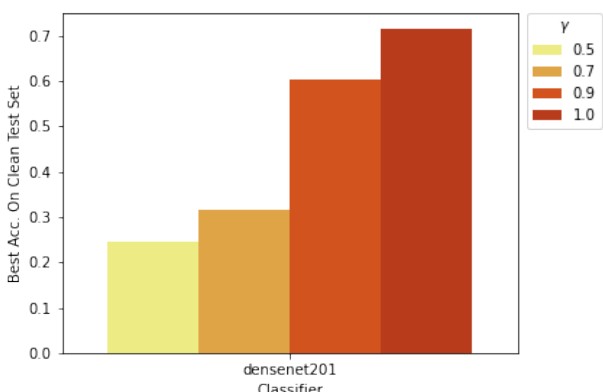

*Figure 13.* Best achievable accuracy after 50 epochs across a range of hyperparameters and random seeds for the brightness modulation approach after applying aggressive training-time augmentations. $\gamma = 1.0$ corresponds to the unmodified, clean dataset for this approach.

*Figure 14.* Examples of perturbed CIFAR-10 data with the pixel-based modification at various settings of the parameter $\mu$.

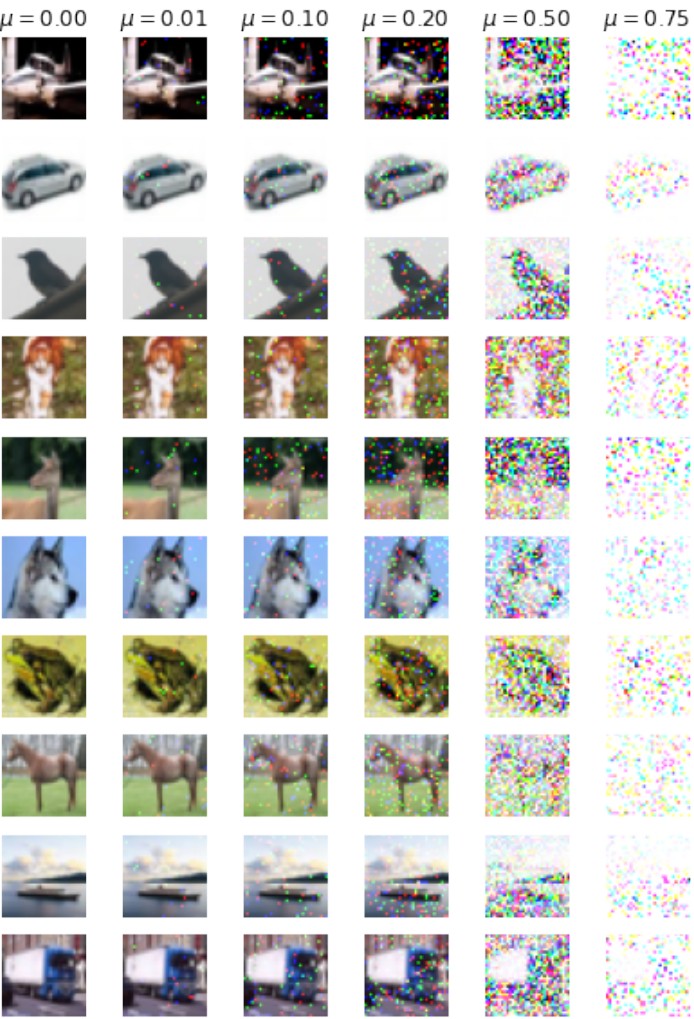

*Figure 15.* Examples of perturbed CIFAR-10 data with the visual watermark modification at various settings of the parameter $\alpha$.

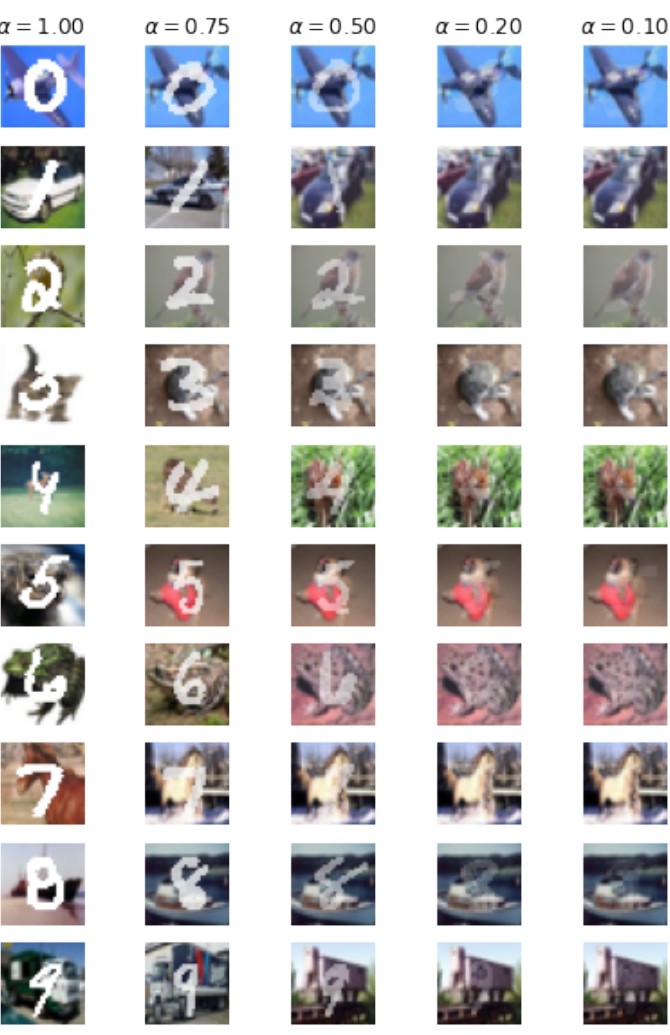

*Figure 16.* Examples of perturbed CIFAR-10 data with the brightness modulation modification at various settings of the parameter $\gamma$.

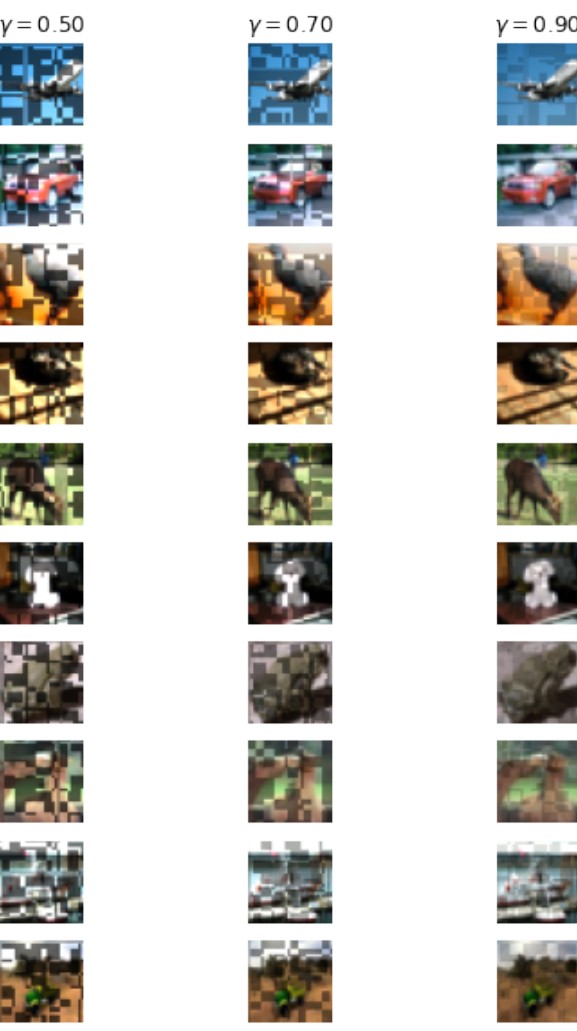