# OpenReview forum: "Disrupting Model Training with Adversarial Shortcuts"
_ICML.cc/2021/Workshop/AML — ICML 2021 Workshop AML Poster_

### Official Review · Reviewer_ujaH · 2021-06-20
**This paper is an application to prevent unauthorized machine learning usage of image datasets which contains 3 kind of simple attack methods.**

**Rating:** Accept
**Confidence:** 4

**Review:**

The paper is clear and fluent.

This paper proposes to apply attack methods to prevent unauthorized machine learning usage of image datasets. It modifies images with sparse pixel-based pattern, visible watermark, or brightness modulation in random rectangles. The modification is unique to each class so as to introduce spurious signals which fool the models to rely on them instead of generalizable features.

The novelty of this paper is limited to applying model attack to data protection. Those modification methods are neither novel nor difficult to remove.

Nevertheless, the experiments are solid to demonstrate that those modification can effectively disrupt model training.

---

### Decision · Program_Chairs · 2021-06-21

**Decision:**

Accept (Poster)

**Comment:**

This paper proposed to apply attacks to prevent unauthorized machine learning usage of image datasets. It provides an interesting application of "adversarial ML for good". The authors can further address the reviewer's comments.